# Detection and Control of Dermatophytosis in Wild European Hedgehogs (*Erinaceus europaeus*) Admitted to a French Wildlife Rehabilitation Centre

**DOI:** 10.3390/jof7020074

**Published:** 2021-01-21

**Authors:** Cécile Le Barzic, Adela Cmokova, Chloé Denaes, Pascal Arné, Vit Hubka, Jacques Guillot, Veronica Risco-Castillo

**Affiliations:** 1Centre Hospitalier Universitaire Vétérinaire de la Faune Sauvage (Chuv-FS), Ecole Nationale Vétérinaire d’Alfort, 94700 Maisons-Alfort, France; cecile.le-barzic@vet-alfort.fr (C.L.B.); chloe.denaes@vet-alfort.fr (C.D.); pascal.arne@vet-alfort.fr (P.A.); 2Department of Botany, Faculty of Science, Charles University, 128 01 Prague, Czech Republic; cmokova@gmail.com (A.C.); vit.hubka@gmail.com (V.H.); 3Institute of Microbiology, Czech Academy of Sciences, 142 20 Prague, Czech Republic; 4Dynamic Research Group UPEC, EnvA, USC Anses, Ecole Nationale Vétérinaire d’Alfort, 94700 Maisons-Alfort, France; jacques.guillot@vet-alfort.fr; 5Laboratory of Mycology, Biopôle Alfort, Ecole Nationale Vétérinaire d’Alfort, 94700 Maisons-Alfort, France

**Keywords:** dermatophytosis, hedgehog, rescue centre, *Trichophyton*, dermatophyte

## Abstract

The rising number of European hedgehogs (*Erinaceus europaeus*) admitted every year to wildlife rehabilitation centres might be a source of concern to animal and public health since transmissible diseases, such as dermatophytosis, can be easily disseminated. This study seeks to evaluate the frequency of dermatophyte detection in hedgehogs admitted to a wildlife rehabilitation centre located near Paris, France, and to assess the risk of contamination in the centre in order to adapt prevention measures. A longitudinal cohort study was performed on 412 hedgehogs hosted at the Wildlife Animal Hospital of the Veterinary College of Alfort from January to December 2016. Animals were sampled once a month for fungal culture. Dermatophyte colonies were obtained from 174 out of 686 skin samples (25.4%). Besides *Trichophyton erinacei*, *Trichophyton mentagrophytes* and *Nannizzia gypsea* were also found. Dermatophyte detection seemed to be associated with the presence of skin lesions, while more than one-third of *T. erinacei*-positive animals were asymptomatic carriers. Healing required several months of treatment with topical and systemic azoles, but dermatophytosis did not seem to reduce the probability of release. Daily disinfection procedures and early detection and treatment of infected and asymptomatic carriers succeeded in limiting dermatophyte transmission between hedgehogs and humans.

## 1. Introduction

*Trichophyton erinacei* is a zoophilic dermatophyte transmitted from hedgehogs; it belongs to the *T. benhamiae* complex, along with eight other zoo- and anthropophilic pathogens [1]. Infection usually occurs by direct contact with an infected hedgehog, although indirect contact with contaminated material such as their nests has also been described [2,3]. While clinical lesions are not always visible in hedgehogs, the symptoms in humans are mostly depicted as inflammatory skin infections [2,3,4]. People at risk of infection are mainly owners of hedgehogs as exotic pets or handlers of these animals in wildlife rescue centres or animal hospitals [5,6]. The general public’s participation in wildlife protection is steadily increasing by means of animal rescue in urban areas or volunteering in wildlife rescue centres [7]. Unfortunately, uninformed intervention in wildlife rescue rises questions concerning the risk of zoonotic disease transmission and the harmonisation of recommendations regarding wildlife handling.

The European hedgehog (*Erinaceus europaeus* Linnaeus, 1758) is a nocturnal insectivorous mammal with a wide distribution throughout western and central Europe [8]. In several regions of its range, the decline of the hedgehog population has been described [9,10]. In 1979, this situation led their inclusion in Appendix III of the Bern Convention on the Conservation of European Wildlife and Natural Habitats. In France, hedgehogs have been designated as a species of conservation concern since 1981 [11]. The presence of *T. erinacei* in European hedgehogs has already been reported [12,13], but, so far, information regarding its prevalence is based on a small dataset. On the other hand, the presence of hedgehog populations as urban dwellers can account for an adaptation to urban and suburban areas because of the loss of their habitat and food sources after urbanisation, agriculture intensification, and the use of pesticides [10,14,15]. Nevertheless, European hedgehogs in urban areas also face a decline in their population that can be associated with several causes, including road collisions [16]. The increasing interaction of hedgehogs with humans stresses the need to establish minimal health precautions to avoid contaminations [17].

In this study, we determine the frequency of dermatophyte detection in injured or orphaned European hedgehogs sheltered at the Wildlife Animal Hospital of the Veterinary College of Alfort, France (Chuv-FS Alfort), over a one-year period. Animals were sampled for the presence of dermatophytes by culture every month until release. Finally, an audit of the protocol for dermatophyte control allowed us to propose safety measures to improve recovery and release times and to avoid cross-contamination between hedgehogs or zoonotic transmission to rescuer workers.

## 2. Materials and Methods

### 2.1. Bioethics

This study was carried out in accordance with the Article 214.90 of the French Rural Code and Directive 2010/63/EC of the European Parliament regarding the protection of animals used for experimental and other scientific purposes. Indeed, this study did not need previous approval by an ethics committee as it did not include any experimental procedure likely to produce pain, suffering, distress or lasting harm equivalent to or higher than that caused by the introduction of a needle, in accordance with good veterinary practices.

### 2.2. Animals and Sampling

All hedgehogs arriving at Chuv-FS Alfort in 2016 were sampled and held complete information regarding origin, age, sex, date of arrival and weight (Appendix A). Before sampling, the animals were submitted to an extensive clinical examination, including the detection of ectoparasites and skin lesions suggesting dermatophyte infection, such as spine loss, crusty skin and erythema [18]. Clinical examination was performed under anaesthesia by isoflurane inhalation (Vetflurane^®^ Virbac, Carros-France). The animals were sampled by scrubbing the whole skin surface with a 5-cm^2^ autoclave-sterilised carpet square [19]. Animals were resampled on a monthly basis until their release. Additionally, the skin scraping test was performed on animals with skin lesions by rubbing off a layer of cells until the capillary oozed with the edge of a blunt scalpel blade. Only dermatophyte-free animals were released.

During the winter season, all dermatophyte-negative hedgehogs arriving in November or later in good body condition (>450 g) were put on hibernation until March in individual boxes located in an isolated shelter. Bedding composed of straw and wood chips, dry food and water were provided when needed, and visual inspections of cages were scheduled twice a week.

### 2.3. Treatment Protocols

Animals with skin lesions suggesting dermatophytosis were treated for one month, four times in a three-day interval, with enilconazole (Imaveral^®^ Audevard, Clichy, France). When dermatophyte colonies were detected by culture, treatment with itraconazole at 10 mg/kg s.i.d (Itrafungol^®^ Elanco, Cuxhaven, Germany) was added to the enilconazole protocol. This treatment consisted of three cycles of one-week treatment with a one-week interval between treatments. Cages and all reusable material in contact with the hedgehogs were washed and disinfected on a daily basis. All materials were cleaned with or immersed in sodium hypochlorite 4% for at least 15 min (La Croix^®^ Colgate-Palmolive Company, Colombes, France).

### 2.4. Culture and Dermatophyte Identification

The skin-exposed carpet face was put in five-second contact with Sabouraud dextrose agar (SDA) plates complemented with 0.5 g/L of chloramphenicol (Merck KGaA, Darmstadt, Germany) and 0.5 g/L of cycloheximide (Actidione^®^, Merck KGaA, Darmstadt, Germany) and then incubated at 30 °C for up to 14 days. All fungal cultures were evaluated both macroscopically and microscopically in terms of colony growth, pigment production and the presence of key characteristics following the morphology keys previously described [20]. Eight selected isolates of *T. erinacei* were subjected to a detailed analysis, which involved the characterisation of micromorphological features and macromorphology on three agar media, including malt extract agar (MEA, Himedia, Mumbai, India), potato dextrose agar (PDA, Himedia, Mumbai, India), and SDA [21] at 25 °C. The macromorphology of the colonies was documented using an Olympus SZ61 or Canon EOS 500D camera, and micromorphology was documented using an Olympus BX-51 microscope. The colour of the colonies was determinate using the ISCC-NBS centroid colour charts [22]. Selected isolates were deposited with the Culture Collection of Fungi (CCF), Department of Botany, Charles University, Prague, Czech Republic.

Confirmation of initial identification from a randomised sampling of 124 positive cultures was performed by sequencing the ITS rDNA region (ITS1-5.8S-ITS2 cluster). DNA was extracted from ten-day-old colonies using the ArchivePure DNA Yeast and Gram2+ Isolation Kit (5 PRIME Inc., Gaithersburg, MD, USA), according to the manufacturer’s instructions, with some modifications [23]. The ITS rDNA region was amplified using the primer set ITS1F and ITS4 [24]. PCR product purification followed the protocol of Réblová et al. [25]. Automated sequencing was performed with Macrogen Sequencing Service (Amsterdam, The Netherlands) using both terminal primers. Editing of the PCR products sequences was performed using BioEdit Sequence Alignment Editor Software [26]. The sequences were compared with those derived from the ex-type and reference strains, which are deposited in the GenBank database of the National Center for Biotechnology Information (NCBI) using the BLAST algorithm.

Furthermore, 16 dermatophyte isolates showing morphological features not compatible with *T. erinacei* were further analysed by mass spectrometry (MALDI-TOF MS). Protein extract samples were obtained, as previously recommended by L’Ollivier et al., for the dermatophytes [27], with slight modifications. Briefly, a small piece of mycelium was gently scraped from the culture plate with a scalpel and suspended in 900 μL absolute ethanol (ethyl alcohol anhydrous; Carlo Erba SDS, Val de Reuil, France) and 300 μL HPLC water (Water HPLC; Prolabo BDH, Fontenay-sous-Bois, France). The sample was vortexed and centrifuged at 13,000× *g* for 10 min, with the resulting pellet air-dried and resuspended in 12.5 μL of 70% formic acid (Sigma-Aldrich, Lyon, France). After 5 min incubation at room temperature, 12.5 μL of 100% acetonitrile (Prolabo BDH, Fontenay-sous-Bois, France) was added over 5 min at room temperature, and the sample was then centrifuged at 13,000× *g* for 2 min. One μL of supernatant was spotted in duplicate onto an MTP 96 target plate polished steel TF (Bruker Daltonics GmbH, Bremen, Germany) and then air-dried. Then, the spot was covered with 1 μL of matrix solution (alpha-cyano-4-hydroxycinnamic acid (Sigma-Aldrich, Lyon, France), saturated in 50:25:25 acetonitrile:HPLC water:10% TFA) and air-dried. A bacterial test standard (Bruker Daltonics) was used for instrument calibration.

MALDI-TOF MS species identification of the spectra was performed using a Microflex LT/SH smart mass spectrometer (Bruker, Bremen, Germany) and the MSI online application (https://biological-mass-spectrometry-identification.com/msi/welcome) developed by Marseille’s Teaching Hospital in collaboration with the BCCM/IHEM collection in Brussels [28].

### 2.5. Statistical Analysis

Animals were categorised into three groups according to their age and weaning status: hoglets (<200 g), juveniles (200–400 g) and adults (>400 g) [29]. Comparisons of dermatophyte detection with origin, sex, age-group, month of arrival, presence of skin lesions and animal outcome (released versus deceased) were made using a chi-square test or Fisher’s exact test when appropriate. The presence of skin lesions and the number of colonies (less than 10 vs. 10 or more colonies) were also analysed using the chi-square test. The length of stay at the rehabilitation centre before release was compared to the detection of dermatophytes by log-rank (Mantel–Cox) test. The same statistical analysis was used to compare the healing time with the number of colonies. All data were analysed using Prism software (v.5, GraphPad software, San Diego, CA, USA). A *p*-value < 0.05 was considered significant. Microconidia and macroconidia sizes were expressed as a size range (mean ± standard deviation).

## 3. Results

Out of the 462 wild hedgehogs rescued during 2016, 412 were included in the present study. The other 50 animals were excluded because three or more epidemiological variables were missing. When data such as age (*n* = 3), sex (*n* = 81) or origin (*n* = 38) were not available during clinical examination, the animal with the missing data was excluded from the corresponding epidemiological analysis.

The origin of animals was mainly from the Ile-de-France region (349/374; 93.3%), but some hedgehogs also came from neighbouring regions such as Centre-Val de Loire, Normandie, Hauts-de-France and Grande-Est (Figure 1). One hedgehog came from as far as the Pays-de-la-Loire region. The number of hedgehogs arriving at the centre fluctuates during the year (Figure 2). Few arrivals occur during the winter season (*n* = 10); later, the numbers increase with time, reaching a peak during the summer season. In 2016, 225 out of 412 hedgehogs arrived between June and August, and half of them were hoglets. In total, 167 of 412 animals were adults (40.5%), 80 out of 412 were juveniles (19.4%) and 162 out of 412 were hoglets (39.3%). The sex rate was homogeneous for the three age groups.

During the clinical examination on arrival, we observed ectoparasites such as fleas (*Archaeopsylla erinacei*) and ticks (*Ixodes* sp.) on most animals, with variable degrees of infestation. The hedgehogs with the poorest body condition on arrival were found systematically infested by Calliphoridae (Diptera) eggs and/or larvae (responsible for cutaneous myiasis). Thirty-two out of 412 hedgehogs (8%) showed skin lesions such as scaly skin, loss of spines or alopecia, with different degrees of severity (Figure 3). Skin scrapings were performed on animals showing itching, crusty skin and erythema on arrival. The presence of mites (*Caparinia tripilis* and *Sarcoptes scabiei*) was observed in two animals, respectively.

### 3.1. Dermatophyte Detection and Identification

Ninety-six out of 412 animals (23.3%) were diagnosed as infected on arrival (Table 1). Cultures revealed the presence of dermatophytes in 186 out of 726 samples (25.6%). *Trichophyton erinacei* was identified morphologically in 174 samples out of 726 (24%). A selection of 124 isolates was submitted for ITS sequencing. The ITS rDNA sequences of 118 *T. erinacei* strains showed 100% similarity with the *T. erinacei* ex-type strain CBS 511.73 (MN974540). Six isolates were identified as *T. mentagrophytes* and showed 99% similarity with the ex-neotype strain of *T. mentagrophytes* IHEM 4268 (MF926358) [30] and 100% similarity with the strain CBS 110.65 (MH858507).

Further identification by mass spectrometry confirmed the identification of 16 isolates as *T. erinacei* (*n* = 10), *Nannizia gypsea* (*n* = 2), *T. interdigitale* (*n* = 2) or *A. quadrifidum* (*n* = 2).

### 3.2. Morphological Characterisation of Dermatophytes

A finely granular, white (#F2F3F4) to light-yellow (#F8DE7E) obverse and a pale-yellow (#F3E5AB) to vivid orange-yellow (#F6A600) reverse characterized colonies of *T. erinacei* on SDA (Figure 4A,B). A coarsely granular, white (#F2F3F4) to yellowish-white (#F0EAD6) obverse and a pale-yellow (#F3E5AB) to vivid orange-yellow (#F6A600) reverse characterized colonies of *T. erinacei* on PDA (Figure 4C,D). A granular, white (#F2F3F4) obverse and a pale-yellow (#F3E5AB) to brilliant-yellow (#FADA5E) reverse characterised colonies of *T. erinacei* on MEA (Figure 4E,F). The colony diameter ranged from 24 to 29 mm (⌀ = 27 mm) on SDA, from 19 to 23 mm (⌀ = 21 mm) on PDA and from 19 to 27 mm (⌀ = 26 mm) on MEA at 25 °C after 7 days. Clavate or pyriform microconidia 2.8–5.5 (4.0 ± 0.8) × 1.6–2.6 (2.1 ± 0.5) μm, borne on short conidiophores, were abundantly present in all samples. Macroconidia were rare, and they usually consisted of 2–4 cells (predominantly two-celled). No spiral hyphae were found even after 21 days of incubation.

Isolates of *T. mentagrophytes* species displayed similar macromorphology to *T. erinacei* on SDA. The main diagnostic criteria to distinguish between these two species are colony reverse on SDA in shades of rusty deep-orange (#BE6516) to deep reddish-orange (#AA381E), coarsely granular colony texture on all examined media (MEA, SDA, PDA) (Figure 5) and the presence of spiral hyphae in most of the *T. mentagrophytes* isolates (Figure 6A). Conidia of *T. mentagrophytes* are predominantly globose to subglobose, while those of *T. erinacei* are predominantly clavate. However, some *T. erinacei* isolates may occasionally show granular colony texture and rusty deep-orange reverse on SDA [31]. Due to the high similarity between these two species and because some isolates did not develop characteristic features, identification was confirmed by sequencing and MALDI-TOF spectra data.

### 3.3. Epidemiological and Therapeutic Analysis

The global rate of dermatophyte-positive samples was 25.4% (174/686), with 169 out of 686 samples being *T. erinacei*-positive (24.6%). This corresponds to 123 out of 412 animals hosted in 2016. Chi-square tests of independence showed no association between the detection of dermatophytes and the origin, sex, age-group or month of arrival (*p* > 0.05). The same proportion of males (38/162; 23%) and females (41/169; 24%) were positive. Estimated trimestrial prevalence of dermatophyte infection or carriage remained at 20–30% during the year. Similar results were observed in hedgehogs classified by age: 39 out of 167 adults (23.4%), 17 out of 80 juveniles (21.3%) and 39 out of 162 hoglets (24.1%) were positive for dermatophytes. Mean prevalence of positive hedgehogs among administrative departments with more than 10 animals was 22.7%, ranging from 13.6% (Val-d’Oise) to 28.6% (Hauts-de-Seine).

On arrival, 20 out of 32 animals with skin lesions (62.5% ) were positive for *T. erinacei*. Among those without skin lesions, 76 out of 380 hedgehogs (20%) were culture-positive. After analysis, there was a significant relationship between these variables—animals with skin lesions were more likely to suffer from dermatophytosis (χ^2^, *p* = 0.0303). Nevertheless, 76 out of 96 positive animals did not show skin lesions (79.2%). Cultures with less than ten colonies per plate were obtained in 55 out 96 of positive animals (57%). Only 20 out of 96 hedgehogs with cultures yielding 10 or more colonies per plate exhibited skin lesions (20.8%) and no statistical association between the number of colonies of dermatophyte and the presence of skin lesions was observed (χ^2^, *p* > 0.05).

The rate of positive animals during their first, second and third sampling was 23.3% (96/412), 32.8% (44/134) and 23.1% (18/78), respectively (Table 1). The contamination by *T. erinacei* at time of arrival was not associated with the outcome of the animals (released vs. deceased). On the other hand, the length of stay was significantly higher (*p* < 0.0001) for positive animals on arrival (mean length stay of 105 days) than for negative animals (mean length stay of 39 days). Finally, 16 hedgehogs with a negative fungal culture were put into hibernation. At the end of the hibernation period, they were sampled again, and 3 out of 16 animals were culture-positive, with 1–5 dermatophyte colonies per plate.

To propose proper safety measures, we screened 41 positive hedgehogs during their entire stay at the centre. Six asymptomatic animals became spontaneously negative one month after their arrival. The remaining 35 animals with skin lesions were further treated. After one month of topical enilconazole treatment, 18 animals (51.4%) became culture-negative or had fewer colonies in their cultures, with a remission of skin lesions. A Mantel–Cox log-rank test confirmed that animals with less than 10 colonies in culture got rid of the infection faster than those with 10 or more colonies (*p* = 0.0016). Following two-month combined therapy (enilconazole plus itraconazole treatment), the remaining 17 hedgehogs (48.6%) were either culture-negative or yielded less than 10 colonies. Among them, eight infected animals became negative for up to 4 months after arrival.

## 4. Discussion

Wildlife care in rescue centres must include the use of appropriate measures to ensure the fast recovery of the hosted animals, avoiding the risk of transmission of infectious agents to other animals or to the caretakers. Our study confirmed that the dermatophyte species *T. erinacei* may be frequently detected in wild European hedgehogs, mostly coming from suburban areas in France, and that this contamination is often unnoticed. A higher prevalence (23.3%) was observed in comparison to a previous study focused on free-ranging European hedgehogs (13%) or captive European hedgehogs (21%) in France [13]. Differences in data may be linked to the inherent causes for their arrival to the wildlife rescue centre (e.g., diseased or injured animals, abandoned hoglets, undernourished animals), which could facilitate dermatophyte infection. High infection rates have been described in urban wild populations in Great Britain and New Zealand, ranging between 20–25% and 44.7%, respectively [12,32].

Previous studies already described no association between the presence of dermatophytes and the observation of evocative skin lesions [12]. The high number of positive cultures coming from asymptomatic hedgehogs (79.2%) confirm that in the context of wildlife rescue, proper measures such as the use of protective gloves and continuous environmental disinfection become crucial to avoid contagion. The monthly survey allowed us to establish close surveillance of the animals with a positive culture, and we could confirm the risk of contamination of the environment and of transmission to other hedgehogs or to handlers [17,33,34]. Previously, Bexton and Nelson [18] highlighted the importance of establishing adequate therapeutic protocols. During our study, topical treatment alone was not enough to cure dermatophyte infection. Indeed, 45.8% of treated animals (44/96) were still infected even after one month of enilconazole treatment. Only combined therapy, i.e., combined topical and oral azole treatment allowed the elimination of the dermatophyte or, at least, a decrease of fungal load in almost all animals (394/412; 95.6%) after up to three months of treatment.

The head of the hedgehogs has been previously described as the most frequent site of infection, while thorough sampling of the animal is also advised since other sites can also be infected [12,35]. We decided to sample animals under isoflurane anaesthesia using the carpet technique, which reduces the risk of underestimating the infection rates [12]. Males and young individuals have been previously described as being more susceptible to dermatophyte infection [12,18]. However, the present study did not reveal any association between fungal load at culture and sex or age. This cannot be explained by the fact that most of the positive animals were asymptomatic, while previous studies were focused on hedgehogs with skin lesions [12,36,37]. Most likely, there would be heterogeneity among the studies to determine the age of hedgehogs, which can only be accurately estimated with a postmortem histological analysis of the jaw [38]. The fact that the prevalence of dermatophytes was not associated with a specific season suggests that environmental conditions may not have a major role in transmission. Nevertheless, a larger number of animals sampled during winter would be needed to confirm this observation, since the risk of contamination through the use of contaminated nests during hibernation has been already described [39]. Finally, the concomitant presence of mites was observed in only two animals of the present study and, as a consequence, it was not possible to confirm the hypothesis that mites may enhance *T. erinacei* transmission, as previously reported by English and Morris [39].

Nest contamination during hibernation could also be an explanation for the three hedgehogs that were highly contaminated at the time of their arrival. They needed three to four months of itraconazole treatment before dermatophyte withdrawal in order to be allowed to hibernate, and they showed a relapse after hibernation. Bexton and Nelson (2016) reported lower cure rates with itraconazole than with terbinafine [18], while recurrence has been described after long-term antifungal treatment with antifungal drugs such as terbinafine [40]. In the present study, the relapse might have happened because the culture technique was not sensitive enough to detect a residual presence of dermatophytes before hibernation, while the straw used as bedding material could have participated in the conservation of remaining dermatophyte spores during the hibernation period. The detection of 21 hedgehogs that were negative at the time of their arrival, becoming positive one month later, seems to support this hypothesis. Further follow-up showed that 16 animals were negative again after one month of local treatment.

Even though the hedgehog-specific dermatophyte *T. erinacei* was the most frequent dermatophyte observed (94.6%), identification by mass spectrometry and sequencing allowed us to confirm the morphological examination as some isolates did not develop characteristic features. Additionally, *T. erinacei* morphology is relatively similar to the second most frequently observed species, *T. mentagrophytes*. While the host spectrum of *T. erinacei* is narrow (Erinaceinae subfamily), *T. mentagrophytes* has been reported from a broad spectrum of domestic animals. However, its host spectrum in wild animals remains poorly known, partly due to extensive taxonomic rearrangements in the past decades [41]. The presence of geophilic dermatophytes *Nannizia gypsea* and *T. terrestre* (syn. *Arthroderma quadrifidum*) in asymptomatic carriers was also reported. These results become of utmost importance if we consider the role of wild animals as carriers of dermatophytes and related fungi [42].

Finally, further research is needed to elucidate whether asymptomatic carriage is due to simple mechanical transport or to infection with isolates with lower virulence, as already suggested for *Microsporum canis* in cats [43]. Low virulence would be associated with decreased keratinolytic activity [44]. The presence of concomitant diseases in dermatophyte-infected hedgehogs does not seem to reduce the probability of release since the rate of positive hedgehogs at time of arrival was higher amongst the animals that were finally released (47/96; 49%) than amongst those that died at the centre (98/316; 31%). A comparative analysis of keratinase and elastase production in *T. erinacei* isolates would shed light on the differential pathogenic risk for hedgehogs and humans.

The high prevalence of asymptomatic carriers detected in this study stresses the risk of dermatophyte dissemination in rescued animals or zoonotic transmission to caretakers. Even if no human contamination was reported during this study, one caretaker and one clinician developed ringworm-associated lesions after the manipulation of animals without protective measures two years earlier. Thanks to the detailed screening of infected animals during their treatment, the study brings up relevant suggestions, such as individual confinement when possible and daily disinfection of cages and tools in contact with the hedgehogs. These measures can be easily be adopted at rescue centres and will ensure the successful treatment of infected.

## Figures and Tables

**Figure 1 jof-07-00074-f001:**
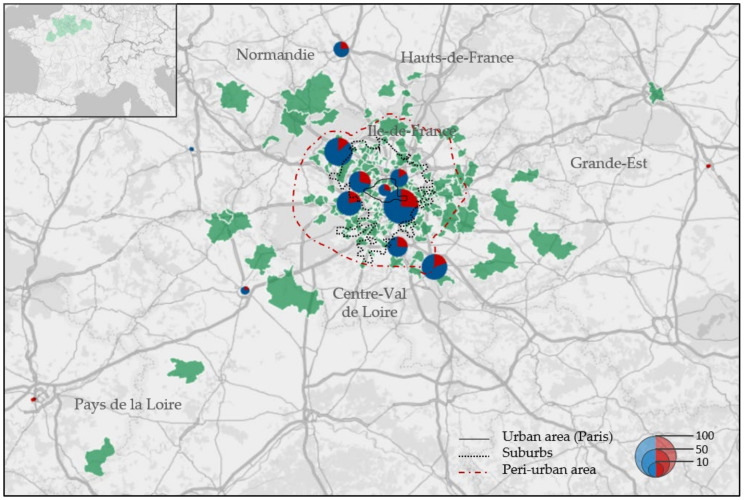
Geographical origin of dermatophyte-negative (blue) and -positive (red) hedgehogs on their arrival at the rescue centre in 2016. Green-coloured regions correspond to departments (upper-left panel) or postal codes (main panel) from where hedgehogs were found. Laboratory results have been grouped by departments. Paris city is included. Size of the circles indicates the number of animals sampled. Administrative and landscaped divisions in Île de France have been adapted from the French Institute of Statistics and Economic Studies (INSEE).

**Figure 2 jof-07-00074-f002:**
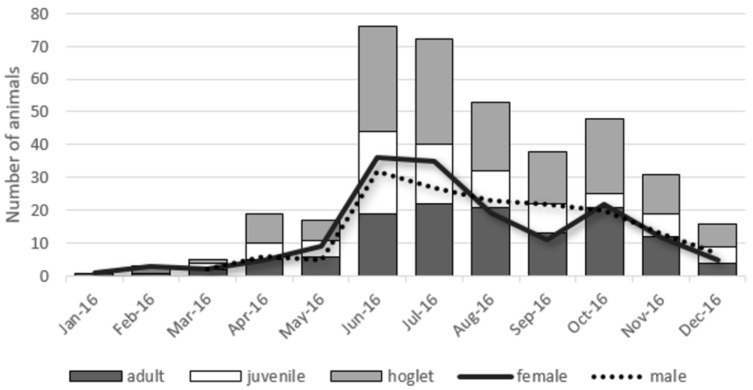
Distribution of hedgehog arrivals over the year according to their age groups (bars) or sex (lines).

**Figure 3 jof-07-00074-f003:**
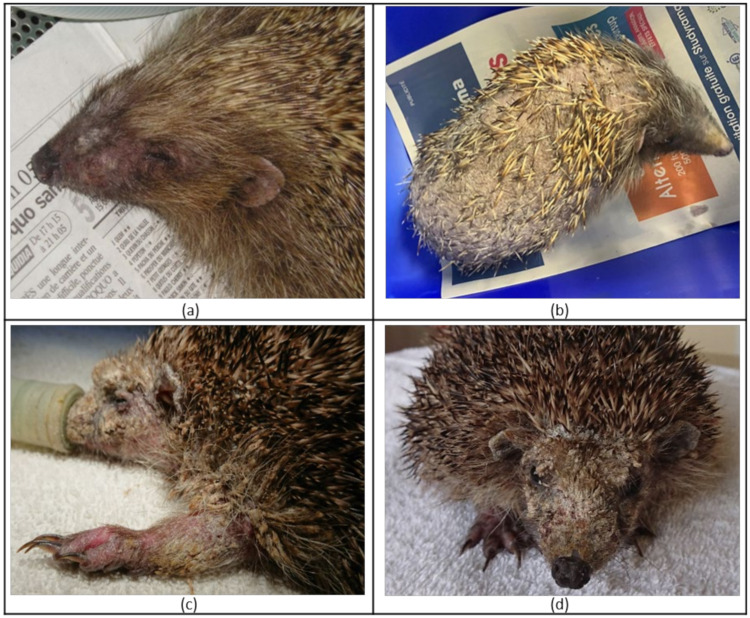
Mild (**a**), moderate (**b**) and severe (**c**,**d**) skin lesions observed in wild European hedgehogs (*E. europaeus*) during clinical examination on admission to the rescue centre. Lesions are often associated with localised (**a**) or generalised (**c**,**d**) scaly skin, alopecia (hair loss), spine loss (**b**) and erythema (**c**).

**Figure 4 jof-07-00074-f004:**
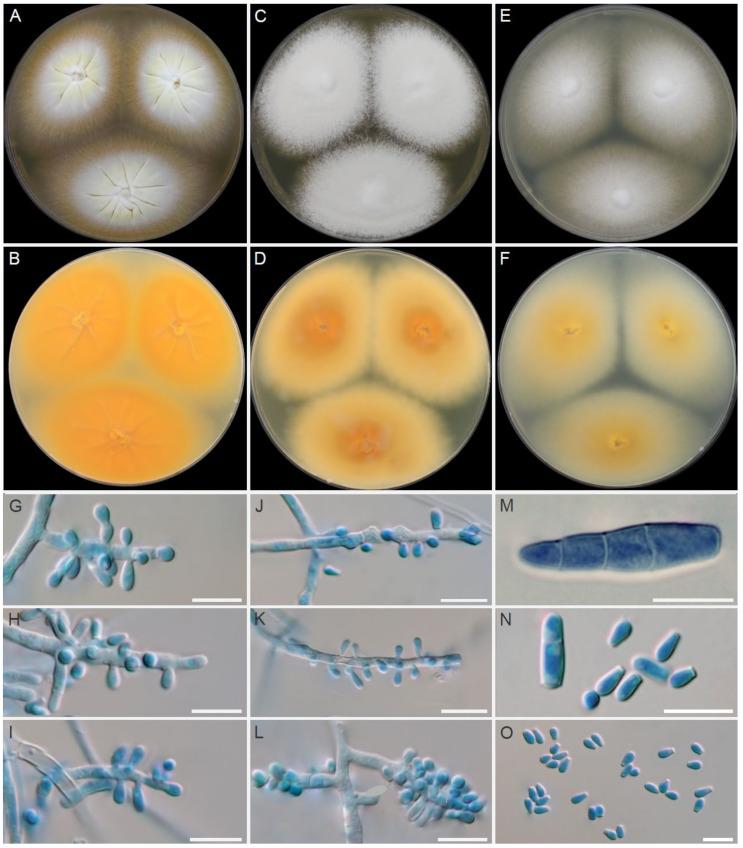
Macromorphology and micromorphology of *Trichophyton erinacei*. Colonies after two weeks of cultivation at 25 °C on Sabouraud’s dextrose agar (**A**,**B**), potato dextrose agar (**C**,**D**) and malt extract agar (**E**,**F**). Conidiophores bearing microconidia (**G**–**L**); macroconidia (**M**); free microconidia and two-celled macroconidia (**N**); microconidia (**O**). Scale bars = 10 μm.

**Figure 5 jof-07-00074-f005:**
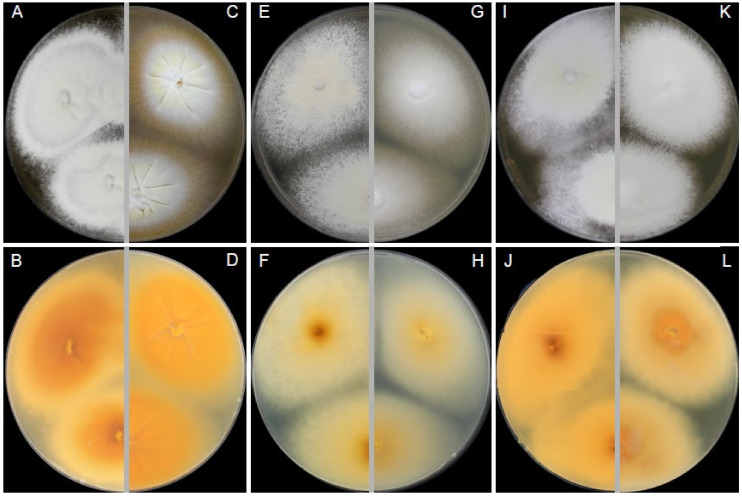
Macromorphology of *Trichophyton mentagrophytes* (left) in comparison with *T. erinacei* (right) isolated from European hedgehogs in France. Colonies of *T. mentagrophytes* after two weeks of cultivation at 25 °C on Sabouraud’s dextrose agar (**A**,**B**), malt extract agar (**E**,**F**) and potato dextrose agar (**I**,**J**). Colonies of *T. erinacei* under the same conditions on Sabouraud’s dextrose agar (**C**,**D**), malt extract agar (**G**,**H**) and potato dextrose agar (**K**,**L**).

**Figure 6 jof-07-00074-f006:**
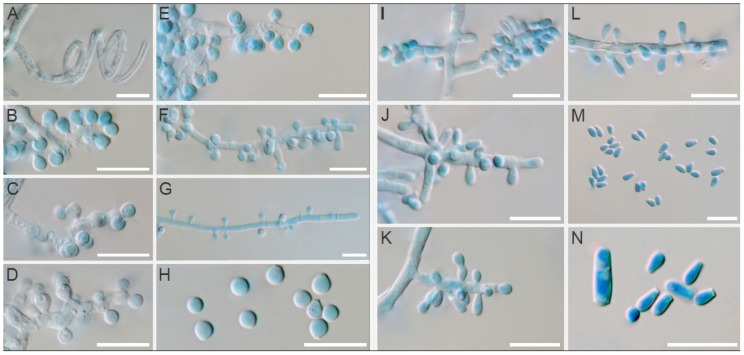
Micromorphology of *Trichophyton mentagrophytes* (left; **A**–**H**) in comparison with *T. erinacei* (right; **I**–**N**) isolated from European hedgehogs in France. Spiral hyphae of *T. mentagrophytes* (**A**); branched or unbranched conidiophores bearing small round microconidia of *T. mentagrophytes* (**B**–**G**); free small round microconidia of *T. mentagrophytes* (**H**); simple conidiophores bearing microconidia of *T. erinacei* (**I**–**L**); free clavate microconidia and two-celled macroconidia of *T. erinacei* (**M**,**N**). Scale bars = 10 μm.

**Table 1 jof-07-00074-t001:** Distribution of culture results over monthly sampling of European hedgehogs admitted to the rescue centre (Chuv-FS Alfort).

Sampling	*T. erinacei*		No Longer Followed (Released/Dead)	TOTAL
	Positive	Negative		
First	96	316	-	412
Second	44	90	278	412
Third	18	60	56	134
Fourth	9	27	42	78
Fifth	1	19	16	36
Sixth	1	4	15	20
Seventh	0	1	4	5
Total	169	517		

## Data Availability

Not applicable.

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
