# Peer review of "Detection and Control of Dermatophytosis in Wild European Hedgehogs (Erinaceus europaeus) Admitted to a French Wildlife Rehabilitation Centre"

_jof, 2021, doi:10.3390/jof7020074_

Round 1

Reviewer 1 Report

Review of the manuscript: “Detection and control of dermatophytosis in wild European hedgehogs (Erinaceus europaeus) admitted in a French wildlife rehabilitation centre”.

Paper presented to the review takes interesting topic on hedgehogs, which were admitted to a wildlife rehabilitation centre from Paris region, France. In the paper the interesting results of studies are presented, in which authors sought to evaluate the frequency of dermatophyte detection in hedgehogs. In my opinion, the manuscript is very well written. I also think, that the methodology is appropriate and sufficient. I must also admit, that in the manuscript many explanations in the text or in descriptions of figures can be found, what makes text easier for reader to understand. However, I had a few comments. I also believe, that after some minor corrections work should be published in Journal of Fungi. My remarks concern the supplementation of the text with the data, which Authors mentioned in the manuscript, but are not included in the figures:

  1. the results confirming general the molecular tests described in this paper, which enabled identification in difficult situations. It would be interesting to see what type of sequences were obtained after sequencing (for example, how long they were), what the sample alignments looked like (especially in cases which were difficult to recognize). Authors also describe MALDI-TOF MS experiments in the manuscript.
  2. information on zoonotic infections among the animal keepers staff

I think, that supplementing the work with these results, that the authors have and which are crucial for the correct identification of strains, would greatly increase the scientific value of the work.

The reviewer declares no conflict of interest.

Author Response

We appreciate reviewer comments and we thank him for his suggestions.

Regarding molecular results, all T. erinacei ITS sequences (559 bp) were the same and matched exactly to the sequence of the strain T. erinacei ex-type strain CBS 511.73 (MN974540). We have not observed any intraspecies variability in ITS. All T. mentagrophytes sequences (678 bp) matched exactly to the sequence of the strain CBS 110.65 (MH858507). There was no variability either. The length of sequences in alignments was as long as the reference gene (ITS) used for blasting for each of the dermatophyte species. We have included information in the manuscript accordingly.

Regarding Maldi-TOF results, As described in material and methods, Maldi TOF analysis was performed by comparing spectra obtained against MSI library of reference spectra.  Decision criteria for MALDI-ToF MS identification was done using a threshold score of 25%. We have modified the corresponding paragraph.

As recommended, we have also added information regarding zoonotic transmission among animal keepers.

We hope all modifications/added information allow the approval of our manuscript for publication

Reviewer 2 Report

The manner in which the results are presented can be improved, particularly the number of positive samples. I did not understand exactly how many samples were positive for Trichophyton erinacei. The morphologic exam detected the presence of T.erinacei in 174 samples and the mass spectrometry added 10 samples to this number = 184. But the number of positive samples presented at the beginning of the "Results" section is 186. It is not clear enough what type of dermatophytes were the other two samples. In Table 1 the total number of positive samples seems to be 169. In the same register, Table 1 would be more clear if a total of positive-negative - not followed samples would be added at the bottom part of it.

Minor revision of the English language and few typos must be corrected.

Examples

Line 242: "futher" -> further

Line 272: replace "to get rid of" with "to cure"

Line 323: replace "to reduce of" with "to reduce"

Line 324 ... animals ( add which)... and amongst (add those)

Line 332-333: Reformulate the last part of the "These measures ... and to successfully treat"

Author Response

We appreciate reviewer comments and we thank him for his suggestions.

Regarding the results, we have modified their presentation as suggested to facilitate its lecture. There was an error when confronting laboratory data and clinical records, and some culture results from excluded animals were included by mistake. In 2016, the laboratory received 686 samples from the 412 animals hosted during 2016. Out of them, 169+4 samples were positive to dermatophytes (174/686; 25.4%). 14 isolates where further identified by Maldi TOF analysis, 2 samples were identified as N gypsea and 2 as T. interdigitale and 6 isolates were confirmed as T. erinacei. 2 negative cultures with suspicious colonies were identified as T. terrestre/A. quadrifidum. Molecular analysis of 124 randomized samples revealed 118 T. erinacei strains showed 100 % similarity with the T. erinacei ex-type strain CBS 511.73 (MN974540) and six isolates were identified as T. mentagrophytes. All this information is now properly described in the manuscript.

Regarding all English corrections, we have revised and corrected grammar and spelling mistakes, plus all suggested corrections.

We hope all modifications/added information allow the approval of our manuscript for publication.

Reviewer 3 Report

A brief summary:

The manuscript titled “Detection and control of dermatophytosis in wild European hedgehogs (Erinaceus europaeus) admitted in a French wildlife rehabilitation centre” by Le Barzic et al. has as main aims to evaluate the prevalence of this zoonotic disease, to identificate dermatophyte using different methods as well as to assess epidemiologic factors and two treatment protocols for dermatophyte control.

The results of this paper are interesting because they help to know better some data of dermatophytosis such as geographic distribution, transmission and the response to different treatments and they corroborate the importance of the environmental disinfection. The finding of high number of T. erinaceid-positive animals without skin lesions (asymptomatic carriers) is an interesting result to establish protocols to control this disease, transmissible to animals and humans.

Broad comments:

Strengths of this study

  • The study was carried out on a high number of skin samples.
  • The use of three methods for the identification and confirmation of different dermatophytes: fungal culture on three agar media (MEA, PDA and SDA), sequencing the ITS rDNA region and mass spectrometry (MALDI-TOF MS).
  • The assessment of therapeutic protocols.

Weaknesses of this study

“Materials and Methods” section:

The authors must add more information in this section to achieve a better understanding of some results, e.g.:

  • How the hibernation of animals was carried out? ¿How many animals?.....
  • Sampling by scrubbing is described but no information on skin scrapings is mentioned in this section.

Description of statistical analysis:

  • The variables and tests used to carry out the statistical analysis are not clearly described in the “Materials and Method” section.
  • Add variables (outcome of animals, treatment protocol, number of colonies of dermatophyte, for example) cited in the “Results” section but not in "Materials and Methods" section. Add also the tests used for their comparison.
  • Indicate as are expressed the results: percentage, mean ± standard deviation….
  • The authors must rewrite better this section.

“References” section:

  • This section needs a deep review.
  • Some data are missing, mainly on books.
  • A reference is duplicated.
  • Some references cited in the text are missing in “References” section.
  • See “Specific comments” below.

This paper is a good work but needed of further revision for the final version.

Specific comments:

Abstract section:

It would be interesting to add some brief conclusions.

Line 27. The word “age” should be changed to “age groups” to avoid confusion.

Line 29. The result cited in the abstract on ”T. erinacei-positive animals, 79.2% were asymptomatic carriers” is missing in “Results” section.

Materials and Methods section:

Line 67. The classification of age groups must be described in this section and deleted in “Results” section (lines 136, 137). Were the criteria for this classification used previously by other authors? In affirmative case, they must include the reference/s in the manuscript.

e.g. Line 67. “…..(Supplementary Data 1). The animals were classified into 3 age groups based on their body weight and weaning status [authors]:…..

Line 74. Add a new section: 2.2. Treatment protocols

Line75. Add the time frame for the administration of enilconazole: “for one month”.

Line 76. The word “complementary” is confusing because it seems that the treatment with itraconazole is administrated together with the first treatment (enilconazole). It needs to be rewritten to clarify the treatment protocol.

Line 78-81. It is not clear if the environmental disinfection was only used together with itraconazole treatment or also along the administration of enilconazole. In line 246, it seems to indicate that environmental disinfection is only applied together with itraconazole but not with enilconazole.

Line 101-102. Two references cited (Gardes & Bruns 1993, White et al 1990) are not in the References section. These references must be included as numbers in a square bracket.

Line 105.  Other reference in text (as http://www.mbio.ncsu.edu/bioedit/bioedit.html) must be included in the “References” section. The authors should refer to the source with a number in a square bracket in the text.

Line 128-132. The authors describe some results obtained by inferential analysis of other variables which are not included in “Material and Methods” section, e.g.:

  • Number of colonies of dermatophytes per plates, <10 and ≥10 (results in lines 232-233).
  • Outcome of the animal (results in line 234).
  • Length of stay (results in lines 235-236).
  • Treatment protocol (results in lines 248-249).

These variables cited in the “Results” should be added in the “2.3. Statistical analysis” section as well as the tests which were used.

Log-rank test is cited in “Results” section (line 245), but this test is not mentioned in the “Statistical analysis” section.

Line 129.  The “body weight” variable measured in g is quantitative and Chi-square test is not appropriate. The same happens with the treatment length variable if it was measured in days or months.

Results section:

Lines 136 and 137. This information on the classification into the age groups should be included in “Material and Methods” section and it should be removed from the “Results” section.

Lines 139-141. It would be interesting to add if the affected regions were rural or urban areas and if the animals were free or captive.

Line 146. I don´t understand “…sex rate displayed a great variability along the year”. I think this variability of the lines which represent female/male in the figure 2 mainly depends on the number de hedgehogs arriving to the centre. These results should be revised and rewritten in the text to achieve a better understanding of the meaning.

Lines 156. Change “prognosis” to “body condition”.

Line 177. Begin other section. The authors must write the text correctly in italics.

Line 179.  Change (Figure 4) to (Figure 4 A, B).

Line 180. Add (Figure 4 E, F) after “…SDA”. What about Figure 4 CD on Potato dextrose agar?

Lines 180-181. Are the results expressed in “mean ± standard deviation” or “standard error of the mean”? It must be indicated. This information is missing also in “Statistical analysis” section.

Line 186. Change “SAB” to “SDA”.

Line 188. Add the letter “A” to the Figure 6 in the text: (Figure 6A).

Line 189. I don´t understand the expression: “this study [27]”? Do the authors refer to “in other study [27]”?

Line 207. Change the letter “H” to “I”.

Line 215. Change the comma of “23,3%” to “23.3%”.

Line 217. Change “age” to “age groups”.

Line 223. Delete the comma in “28.6,%”.

Lines 235-236. The authors give the mean of length of stay (in days) but the mean is expressed as mean ± standard deviation. As mentioned above, this quantitative variable and the test used to compare the means are not cited in “Materials and Methods”.

Line 242. I think it is better to make emphasis on that “the remaining 35 animals” had skin lesions. e.g. “The remaining 35 animals with skin lesions were…..”

Line 243. “51,4%” instead of “51%”. Other percentages must be revised.

Discussion section.

Line 273. The authors must change (44/99) to (44/96).

Lines 277-278. Rewrite the sentence to clarify that the authors carried out the thoroughly sampling of the animals due to other sites can also be infected as has been described in other studies [11,31].

Line 280. The statement of “reduce the risk to underestimate the infection rates” needs a reference [author].

Lines 282-284. I don´t understand as no association between fungal load and sex or age can be explained by the fact that most of the positive animals were asymptomatic in the present study unlike others.

Line 289. The expression “risk of contamination” should be changed to “risk of infection”.

Line 293. The reference [36] must be changed to [35]. In the “References” section, the [35] and [36] are the same reference. It is duplicated and the following references need to be renumbered.

References section:

This section shows a lot of mistakes and it needs a deep review. The authors must modify this section based on “Reference List and Citations Style Guide for MDPI Journals”.

Cited journals should be abbreviated according to ISO 4 rules and references to books should cite the author(s), title, publisher, publisher location (city and country), publication year, and page, but the authors did not always follow these rules. e.g.:

  • References of journals which are not abbreviated: number 4,6,11, 14, 17, 18, 24, 25, 28, 29, 31, 34, 38.
  • Title is missing in the reference number 5
  • Journal is missing in the reference number 9
  • Book data are incomplete: reference number 7, 19, 20,21….
  • References 35 and 36 are the same.
  • In the reference [40], the authors have to modify the title: “…Microsporum canis and the cat” must be changed to “Microsporum canis in the cat”.

Species name is Italicize, but the authors have not followed this rule in the “Reference” section.

The authors must look for other errors and modify them.

Figures:

Figure 1.  It would be interesting to add more information, e.g.:

  • The authors could indicate the urban area and, if is appropriate, the rural area in the map, delimiting these areas within circles.
  • The location of Pays de la Loire and Marne should be added in the small map of France located in the upper left quadrant.

Figure 2. The authors should change the color of the bars and lines to improve the visualization of the results.

At the bottom of the “Figure 2”, the authors should change “age” to “age groups” to indicate that the bars represent a categorical variable but not a quantitative one.

Figure 3. The erythema is more evident in picture (c) than in picture (d). Is alopecia and spine loss the same lesion? It is not clear in the text at the bottom of the “Figure 3”.

Figure 4. Some letters of “Figure 4 (G-O)” are not clearly visible because they are white in a clear background.

Supplementary Data.

  • Add the number 1: “Supplementary Data 1”, as the authors cited in the text of the manuscript (line 67).
  • In column “Outcome”, the initials (E, R, D and NA) must be identified to know their meaning.
  • The column “Weight at arrival (g)” includes quantitative data. For this reason, the words “juvenile” or “adult” should be deleted in this column.
  • The authors should add another column in the table which includes an “age group” variable.
  • There are 2 empty cells in the column “outcome” (number 269 and 386). What is the reason?

Author Response

We appreciate reviewer comments and we thank him for his suggestions.

Material and methods section.

We have included a paragraph regarding hibernation procedure and we have explained the skin scrapping method used.

We have modified and improved the statistical analysis section, as suggested

References section

We have verified our library software (Zotero v. 5.0.94) uses Journal of Fungi style of citation, and we have manually corrected all imported references according to ISO 4 rules. We have corrected and adapted all references according to JoF rules.

Specific comments: we have modified/added information as suggested regarding the introduction, the material and methods and the results sections.

We hope all modifications/added information allow the approval of our manuscript for publication
